# Properties of Leukemic Stem Cells in Regulating Drug Resistance in Acute and Chronic Myeloid Leukemias

**DOI:** 10.3390/biomedicines10081841

**Published:** 2022-07-30

**Authors:** Xingjian Zhai, Xiaoyan Jiang

**Affiliations:** 1Terry Fox Laboratory, British Columbia Cancer Research Institute, Vancouver, BC V5Z 1L3, Canada; xzhai@bccrc.ca; 2Experimental Medicine, Department of Medicine, University of British Columbia, Vancouver, BC V5Z 1M9, Canada; 3Department of Medical Genetics, University of British Columbia, Vancouver, BC V6H 3N1, Canada

**Keywords:** leukemic stem cells, acute myeloid leukemia, chronic myeloid leukemia, multi-omics, drug resistance, metabolism, combination therapy, signal transduction, tumor microenvironment

## Abstract

Notoriously known for their capacity to reconstitute hematological malignancies in vivo, leukemic stem cells (LSCs) represent key drivers of therapeutic resistance and disease relapse, posing as a major medical dilemma. Despite having low abundance in the bulk leukemic population, LSCs have developed unique molecular dependencies and intricate signaling networks to enable self-renewal, quiescence, and drug resistance. To illustrate the multi-dimensional landscape of LSC-mediated leukemogenesis, in this review, we present phenotypical characteristics of LSCs, address the LSC-associated leukemic stromal microenvironment, highlight molecular aberrations that occur in the transcriptome, epigenome, proteome, and metabolome of LSCs, and showcase promising novel therapeutic strategies that potentially target the molecular vulnerabilities of LSCs.

## 1. The Discovery and Cellular Properties of Leukemic Stem Cells (LSCs)

### 1.1. Historical Perspectives of LSCs

Stem cell biology was a boldly novel discipline in the early 20th century, beginning with the proof of existence of bone marrow (BM) hematopoietic stem cells (HSCs) by James Till and Ernest McCulloch in the 1960s after successfully reconstituting blood cells on heavily irradiated mice [1]. On a societal level, at the time, Till and McCulloch’s discovery also contributed to the theoretical framework supporting the clinical implementation of BM transplantation for those exposed to severe nuclear radiation during the Cold War [2]. In the decades that followed, expansion of cellular and molecular characterization of the physiological properties of HSCs scientifically and militarily led to the establishment of an intricate blood cell differentiation hierarchy [3] and, more importantly, the development of technologies such as fluorescence-activated cell sorting [4] and advances in tissue culture assays, including the long-term culture-initiating-cell (LTC-IC) assay that allows the differential clonal output potential of single HSCs to be examined; these all accelerated HSC research [3,5].

About 30 years after the discovery of HSCs, in the context of oncogenesis, the concept of the existence of a select subpopulation of leukemic cells not only capable of sustained self-renewal but also able to repopulate and give rise to human leukemias was demonstrated in vivo in acute myeloid leukemia (AML) by John Dick and colleagues [6,7], as well as in chronic myeloid leukemia (CML) by Connie Eaves and colleagues [8,9,10]. These cells with strong leukemia-initiating capacity, termed leukemic stem cells (LSCs), are found to be immunophenotypically enriched within the Lin^−^CD34^+^CD38^−^ BM cell fraction, despite the high degree of surface and intracellular marker heterogeneity therein [11,12,13]. The experimental basis for such seminal discoveries involved using limiting dilution of fractionated cell pools harboring distinct surface markers, followed by engraftment into immunocompromised murine hosts. These techniques, though time-consuming, effectively isolated cell population enriched for LSCs, allowed relative quantification of LSC frequency, and even served as a contemporary in vivo gold-standard approach for functional validation of LSC activity [14]. With the advent of next-generation sequencing, high-throughput multi-omics technologies, and sophisticated murine models developed in the 21st century, scientists are now able to further interrogate the nature and evolutionary trajectory of LSCs, sprouting the seed for an era of advanced stem cell research. Indeed, recent mounting evidence corroborates that LSCs are heavily involved in refractory hematological malignancies, particularly AML and CML. In this review, we aim to provide a conceptual framework for the phenotypical, cellular, and molecular features that enable LSCs to survive and persist in AML and CML patients, all the while highlighting several potential therapeutic avenues to target the distinctive molecular vulnerabilities of these LSCs, with the overarching goal of eradicating residual malignancies.

### 1.2. Surface Antigens of AML LSCs

Despite recent advancements made in characterizing LSCs, it is worth noting that LSCs should not be perceived to have well-defined, uniform biomarkers. Rather, LSC surface markers can be fluidic and even context-dependent. Dick and colleagues’ seminal discovery in 1997 uncovered that cells expressing surface markers CD34^+^CD38^−^ were able to differentiate in vivo in severe-combined-immunodeficiency-disease (SCID) mice into leukemic AML blasts; however, this description was broad and preliminary, as studies later revealed this same set of surface marker is also shared by normal HSCs and progenitor cells, sparking interest in the search of cellular markers that specifically encapsulate LSCs [15,16,17]. Moreover, early characterization of AML LSCs was predominantly antibody-based; however, such practices may lead to biases resulting from differing reactivities and specificities of the antibodies used and the harsh assay environment associated with cell sorting/fractionation, as not all cell types could be sorted with the same efficiency in vitro and in vivo [18,19]. Nowadays, recognizing these potential pitfalls, research efforts have concentrated on examining surface antigens that appear to be aberrantly regulated or differentially expressed on LSCs as opposed to their healthy stem and progenitor cell counterparts. One promising marker is CD33, which is highly expressed in AML patients, and its expression along with T cell immunoglobulin and mucin protein (TIM3) specifically denotes AML cells as opposed to normal hematopoietic tissues [20,21]. Another marker that is reported to be preferentially overexpressed in CD34^+^CD38^−^ AML cells is CD123, and its expression serves as a clinical marker for adverse patient outcome; it is potentially mediated by mechanisms involving STAT5 activation [22,23,24,25]. Furthermore, CD47, which upon interaction with SIRPα on circulating macrophages and dendritic cells inhibits phagocytosis to facilitate immune evasion of AML cells, is overexpressed in CD34^+^CD38^−^ LSCs, conferring a survival advantage to AML stem cells [26,27,28]. Using signal-sequence-trap technology to identify surface antigen expression milieu, followed by quantitative real-time polymerase chain reaction (qRT-PCR) validation across Lin^−^CD34^+^CD38^−^ AML samples and normal BM-derived HSC/progenitor cells, Hosen and coworkers found CD96 to be preferentially upregulated in AML cells and demonstrated LSC activity among CD34^+^CD38^−^CD96^+^ AML cells through successful engraftment of these cells into irradiated immunocompromised mice [29]. Furthermore, transcriptomic analysis done on purified HSCs from myelodysplastic syndrome (MDS) patients and age-matched cord blood cell HSC controls identified selective overexpression of CD99 on LSCs, and that CD99 is at a particularly high level at relapse when compared to samples obtained at time of diagnosis, suggesting a potential link between CD99 expression and the chemo-resistant properties of LSCs in general [30]. Furthermore, a study conducted by Heo and colleagues reported prominent enrichment of the CD45^dim^CD34^+^CD38^−^CD133^+^ cells in BM of AML patients, and that this unique signature is associated with poor prognosis; however, functional validation of LSC activities of these cells is still needed [31]. Another factor that complicates the already heterogenous landscape of AML LSC surface antigen expression is the stage during which leukemic transformation occurs, as recent evidence reported transformation may also take place in mature granulocyte–macrophage precursors without expressing CD34, specifically in those samples with *NPM1* mutation [32,33,34]. In other instances, leukemic-initiating cells are found to exist in the CD38^+^ progenitor cell population [35]. These novel insights may challenge the way in which we purify and interpret the surface antigen milieu of AML LSCs. In a nutshell, AML LSCs harbor an extremely complex surface antigen expression profile, and further consensus is needed to define these molecular markers. Of equal importance is the mechanistic understanding underlying such surface markers in relation to stemness and clinical drug resistance. 

### 1.3. Surface Antigens of CML LSCs

The first evidence of a highly primitive type of leukemic cells present in CML patients was reported in 1992 using techniques such as LTC-IC assay followed by limiting dilution assays of patient CML blood and BM samples [36,37]. A later study published in 1999 from the same research group established the quiescent nature of CML LSCs after isolating a quiescent cell fraction among CD34^+^ CML LSCs using nucleic-acid-binding agents Hoechst 33342 and Pyronin Y, and subsequently confirming the engraftment capability of these quiescent CML LSCs on immunodeficient mice [38]. This dormant state was later discovered to be an important feature of drug-resistant CML LSCs, as they contribute to the root cause of drug resistance and disease relapse in CML patients [39,40]. Since then, many studies have aimed to characterize surface biomarkers selectively present on CML LSCs; however, just like the dilemmas encountered in defining the surface marker landscape of AML LSCs, evidence in CML LSCs appears to be controversial due, at least in part, to the lack of standardized in vitro and in vivo techniques employed and the molecular heterogeneity intrinsic to the functioning of CML LSCs throughout different stages of disease progression [41]. Nevertheless, studies to date have mainly focused on examining the surface marker expression of CD34^+^CD38^−^ CML cells in the chronic phase, a comparatively stable stage of CML pathogenesis. According to Herrmann and coworkers, CD34^+^CD38^−^ CML patient cells in the chronic phase are reported to have a roughly 10-fold higher expression of CD33 compared to normal CD34^+^CD38^−^ stem cells by flow cytometric analysis, even though CD33 remains comparable at the transcript level [42]. Interestingly, IL-1 receptor accessory protein (IL1RAP), which is found to be upregulated in CD34^+^ and CD34^+^CD38^−^ CML cells, appears to increase as patients transition from the chronic phase to accelerated and blast crisis phases [43]. Furthermore, using gene array, qPCR, and flow cytometric analyses, Hermann et al. reported that CD34^+^CD38^−^CD26^+^ but not CD34^+^CD38^−^CD26^−^ LSCs obtained from chronic-phase CML patients contained *BCR/ABL1* mRNA, exhibited repopulating capability in NSG mice, and were highly expressed in imatinib-nonresponder patients, suggesting that CD26 is a highly specific CML LSC biomarker and a potential therapeutic target [44,45]. Complementing this line of evidence, a more recent study using single-cell gene expression analysis in conjunction with immunophenotypic screening revealed a distinct signature of Lin-CD34^+^CD38^−/low^CD45RA^−^cKIT^−^CD26^+^ as being associated with a particular CML LSC compartment that is relatively insensitive to tyrosine kinase inhibitor (TKI) treatment [46]. Furthermore, a study published in 2020 comparing a total of 878 BM or blood samples from 274 patients with AML, 97 patients with CML, and 288 controls also revealed an aberrant profile of CD25^+^CD26^+^/CD56^+^/CD93^+^/IL1RAP^+^ antigens among CD34^+^CD38^−^ CML cells [11]. Indeed, characterization of the surface marker landscape of CML LSCs not only aids in the phenotypic purification of these intriguing cells, but also paves the way for the identification of specific, actionable, and prognostic biomarkers on CML LSCs for therapeutic design and prevention of disease relapses. As exciting as these discoveries may seem, further studies are still needed to identify CML LSC surface antigens potentially driving disease progression from chronic to accelerated and blast crisis phases and/or conferring TKI resistance. A concise summary of AML and CML LSC surface antigens is provided in Table 1.

### 1.4. Clinical Challenges of LSCs 

Owing to their unique molecular properties, including self-renewal, phenotypic plasticity, and the wondrous ability to differentiate and repopulate new tissues, LSCs are highly versatile entities, insidiously perpetuating therapeutic resistance [47,48]. Indeed, genetic variants known to evade therapeutic treatments are enriched in stem-like cells found across almost all forms of leukemia [49,50,51,52,53]. Among the molecular aberrations present in cancer stem cells, drug-induced senescence and quiescence represent predominant causes of patient relapse, as most antineoplastics target processes associated with actively dividing bulk cancer cells, not those in the quiescent state [54,55]. As LSCs represent just a rare fraction of the total bulk cancer cells, patients undergoing chemotherapy can be considered to have reached therapeutic endpoints when, in fact, those LSCs may still remain alive. More so, several groups have also reported that cancer stem cells can become “trained” to respond to chemotherapeutic insults either through crosstalk with the tumor microenvironment or acquisition of adaptive genomic circuitry to mitigate drug-mediated cytotoxicity or to bypass drug-targeted pathways, fortifying their ability to withstand therapeutic challenges [56,57,58]. This creates a clinical dilemma where drug resistance and relapses are not only permitted, but are sometimes even inadvertently encouraged once patients go through round after round of therapy. These complications, coupled with the heterogenous surface marker expression of LSCs, nonspecific cytotoxicity of existing antineoplastics, and inter-patient sensitivity to standardized treatments, collectively stagnate the overall survival of leukemia patients, especially that of elderly individuals. 

## 2. Cell-Extrinsic Signaling: LSC Interaction with the BM Microenvironment

### 2.1. LSC BM Homing and Mobilization

Stem cell homing describes the process of circulating stem cells migrating to an environmental niche, typically within the BM [59]. The BM, in turn, offers a cytoprotective microenvironment in which LSCs are generally shielded from cytotoxic insults, forming the basis for therapeutic resistance and stable engraftment [60]. As a result, understanding the factors that regulate the trafficking and localization of LSCs to the BM is critical to understanding LSC-mediated drug resistance [61]. It is known that stromal ligand binding to LSC surface receptors not only provides physical anchorage for circulating LSCs to the BM niche but may also activate oncogenic intracellular signaling pathways. For instance, in CML, dissociating the interaction between E-selectin, an adhesion molecule expressed on the endothelial cells within the endosteal BM microenvironment, and CD44 using the E-selectin inhibitor GMI-1271 sabotages BM homing of LSC-enriched CML cells, enhancing CML sensitivity to eradication by TKIs such as imatinib mesylate and prolonging survival of mice with CML [62,63]. In line with this finding in the context of AML, E-selectin binding to AML blasts activates the pro-survival AKT/NF-κB pathway, conferring AML resistance to cytarabine (Ara-C) [64]. Another signaling axis that exemplifies the duality of LSC BM homing and activation of downstream survival pathways, while also having been characterized preclinically in the context of AML and B-cell ALL chemoresistance, is the interaction between chemokine receptor 4 (CXCR4) and stromal cell-derived factor 1 (SDF-1 or CXCL12) found on BM endothelial and stromal cells [65,66]. On the contrary, a subsequent study by Ramakrishnan et al. demonstrated the dispensability of CXCR4 in BM homing of stem-cell enriched MLL-AF9 AML cells in sub-lethally irradiated mice, despite the critical need of CXCR4 for growth by AML cells in said model [67]. Yet another study reported that deletion of *CXCL12* in mesenchymal stromal cells resulted in CML LSC expansion and increased sensitivity to TKI treatment, whereas endothelial cell-specific *CXCL12* deletion decreased LSC proliferation [58]. These results point towards context-dependent, differential regulation of the signaling interface between LSCs present in various leukemias and their respective BM niches. Additionally, upon interacting with fibronectin on stromal cells, high levels of α_4_β_1_ integrin, besides enabling HSC retention in the BM, mediate chemoresistance in AML cells via the PI3K/AKT/BCL-2 signaling pathway and are often associated with BM minimal residual disease [68,69]. The significance of other integrin family members such as β_3_ integrin and α_6_ in stromal preservation and maintenance of LSCs is also documented in CML [70] and B-ALL [71], respectively.

Since BM homing of LSCs is known to confer drug resistance, then triggering LSC release from the BM niche may represent a promising therapeutic strategy. LSC mobilization can be achieved through endogenous proteases such as neutrophil elastase (NE), cathepsin G (CG), and matrix metalloproteinase 9 (MMP9) cleaving membrane-bound adhesion complexes, or therapeutically with small-molecule drugs either by directly rewiring the broader cytokine milieu involved in LSC and HSC BM homing or specifically antagonizing the BM homing signaling pathway, such as CXCR4 and CXCL12 [72,73]. Promisingly, several small-molecule inhibitors are successful against various malignancies, including AML, multiple myeloma, and non-Hodgkin’s lymphoma, and are also administered pre-operatively to maximize peripheral blood HSCs prior to autologous or allogenic HSC transplantation [74]. This evidence highlights the dynamic interplay of soluble factors and the cell adhesion network that facilitates BM homing and engraftment of LSCs, and how the disruption of such processes may help eradicate repopulating LSCs. 

### 2.2. Stromal Cellular Signaling in LSC-Mediated Drug Resistance

Surrounded by a constellation of extracellular matrix (ECM) components, the majority of LSCs can generally be found in the endosteal (close to bone) and perivascular (close to sinusoids) compartments within the BM cavity [75,76]. LSCs engage in bidirectional crosstalk with a variety of cell types, including osteoblasts, osteoclasts, mesenchymal stromal cells, endothelial cells, neutrophils, and adipocytes in order to establish a malignant environment that supports the survival, unlimited self-renewal capacity, and quiescent nature of LSCs [77,78,79]. Several selected key constituents of the stromal microenvironment and their reciprocal interactions with leukemic cells are discussed below (Figure 1). It is worth noting that to remodel or create a malignant niche conducive to the survival, quiescence, and drug-resistant properties of LSCs, multiple cell types, outlined in the following subsections, are known to dynamically interact with one another, oftentimes coupled with other molecular aberrations including Wnt overactivation [80], Dicer 1 deletion [81], and release of BM-modeling leukemic exosomes by leukemic cells, per se [82]. Taken together, the BM leukemic niche is not only capable of sustaining LSC persistence, hence playing a direct role in malignant transformation, but also remains prone to malignant remodeling by LSCs and leukemic blasts.

#### 2.2.1. Osteoblasts

Recent studies have reported conflicting roles of osteoblasts residing in the BM endosteal niche in the maintenance of HSC and LSC pools, potentially due to the varying underlying BM microenvironments across several leukemias. Conditional osteoblastic ablation in mice results in loss of BM cellularity and progressive loss of BM HSCs [83]. Since osteoblasts appear to support normal hematopoiesis, an increased osteoblastic number suppresses leukemic burden in mice with ALL [84]. In CML, however, osteoblasts are reported to confer niche-mediated LSC resistance against TKIs, and the dual PI3K and mTOR inhibitor BEZ235, which inhibits osteoblastic and endothelial cells in the BM microenvironment, enhances sensitivity of primary CML LSCs to TKI therapy [85]. On the contrary, parathyroid hormones secreted from osteoblasts act to decrease CML LSCs but differentially upregulate AML LSCs [86,87]. Nevertheless, it appears that both AML and CML LSCs crosstalk with osteoblasts to foster a pro-inflammatory tumor microenvironment [88,89,90]. These studies suggest that BM constituents often exert non-unilateral effects during leukemic progression. Perhaps equally importantly, consideration must be given when interpreting these studies, as limitations range from the potential pleiotropic effects of using small-molecule inhibitors, upon which several studies are based, to unideal culturing conditions that may compromise osteoblastic growth due to failure to comprehensively assimilate the dynamic BM microenvironment during the study.

#### 2.2.2. Mesenchymal Stromal Cells

Physical contact with leukemic cells seems to be key for mesenchymal stromal cell (MSC)-mediated BM microenvironment modulation. In a direct-contact co-culture system with primary AML patient cells, MSCs protect AML cells from Ara-C cytotoxicity and enrich the quiescent population compared to media control alone [91]. To account for the cytoprotective effect of MSCs, MSCs are known to physically associate with AML and T-ALL leukemic cells via tunneling nanotubes, within which mitochondria are transferred bilaterally in an attempt to repair mitochondrial dynamics post-chemotherapeutic exposure [51,92,93]. Mechanistically, MSC-mediated AML chemoresistance requires gap-junction constituent connexins, and the use of a gap junction disruptor, Carbenoxolone, elicits synergistically antileukemic killing with Ara-C in vitro and in vivo [94]. Furthermore, in CML, stromal cells appear to play a critical role in maintaining LSC dormancy partly through the bone morphogenetic protein 4 (BMP4)-BMP receptor 1B (BMPR1B) pathway, cooperating with JAK2/STAT3 signaling to induce quiescent gene expression and lead to residual LSC persistence [95]. Indeed, the correlation between physical adherence to the stroma and the acquisition of stem-like features and subsequent drug resistance has also been documented in B-ALL and AML, illustrating an intimate and protective niche created by MSCs for LSCs [69,96].

#### 2.2.3. Endothelial Cells

Located in the perivascular compartment of the BM cavity, endothelial cells regulate multilineage hematopoiesis through the secretion of soluble factors and protect vascular integrity against inflammatory insults [97,98,99]. Many studies have collectively portrayed endothelial cells as angiogenic facilitators, hence promoting leukemogenesis, but characterization of such leukemogenicity through the lens of LSCs remains insufficient [100,101,102]. A clever avenue through which vascular endothelial cells crosstalk with LSCs is through the release of microRNAs (miRNAs). For instance, miR-126 has been known to maintain the quiescent phenotype of LSCs [103,104]. In CML, stromal endothelial cells may further enrich this quiescent phenotype of LSCs by transferring miR-126 to CD34^+^ CML cells via extracellular vesicles, and the combination of TKI (nilotinib) and an inhibitor of miR-126 (CpG-miR-126) enhances in vivo targeting of CML LSCs [105]. Quite extraordinarily, AML cells have been reported to integrate into the vasculature, becoming vascular tissue-associated AML cells and adopting a quiescent phenotype while retaining the ability to give rise to AML upon transplantation [106].

#### 2.2.4. Adipocytes

Adipose tissue (AT) is a critical tumor microenvironment modulator, mainly through its endocrine function of regulating hormones, cytokines, and soluble factors capable of interacting with diverse immune cells [107,108,109]. Since tumor cells often adopt distinct metabolic profiles from their healthy counterparts as they navigate and survive in their tumor microenvironment, adipocytes in ALL, for instance, supply leukemic cells with unsaturated free fatty acids that, during nutrient deprivation, can be utilized by ALL cells for β-oxidation [110]. In line with this finding, adipocytes have been reported to serve as a reservoir for CML LSCs, particularly those with high expression of the fatty acid transporter CD36 and that are metabolically adapted to benefit from fatty acid oxidation, to hide and thrive in an extramedullary AT microenvironment to evade chemotherapeutic challenge [111,112]. With adipocytes providing an alternative supportive niche in which LSCs survive, these findings may potentially account for, at least in part, the clinical observation that links obesity to generally poorer prognosis in patients with hematological malignancies [113].

## 3. Cell-Intrinsic Signaling: Aberrant Multi-Omics Circuitry of LSCs in Drug Resistance

### 3.1. Induction of LSCs from HSCs through Pre-Leukemic Stem Cells

Pre-leukemic stem cells (pre-LSCs) are considered to have originated from sequential oncogenic transformations of healthy HSCs. These resultant pre-LSCs are resistant to chemotherapeutics and serve as promising players in disease relapse [114,115]. Many initiating mutations begin in prominent epigenetic modifiers, such as DNA methyltransferase 3A (*DNMT3A*) mutations followed by those of ten-eleven translocation 2 (*TET2*) [114]. Interestingly, these early-stage mutations, such as *DNMT3A* mutations, reinforce stemness phenotype, further enhancing leukemic progression and LSC formation [116]. Upon undergoing further cytogenetic mutations, pre-LSCs eventually adopt LSC-associated molecular signatures, silently driving leukemogenesis with their endless self-renewal capacity and quiescence state. Owing to the high degree of semblance between pre-LSCs and HSCs, some even term pre-LSCs as preleukemic HSCs. As a result, it is unsurprising that conventional surface-antigen-based cell-pool fractionation may not be sufficient to phenotypically purify pre-LSCs from HSCs and LSCs, especially considering that these three cell cohorts differ mostly in terms of gene expression patterns, mutational status, and differentiation states. Nevertheless, more-sensitive techniques involving a strategic combination of single-cell transcriptomics with lineage tracing based on nuclear and mitochondrial genetic variants has shown promise in distinguishing pre-LSCs from their LSC and HSC counterparts [117]. Additionally, the majority of pre-LSC studies are done in the context of AML, leaving many unknowns in CML. Another limitation is that even in well-characterized AML pre-LSC studies, there is a lack of a precise, clinically characterized latency period in which HSCs acquire and adopt stable pre-LSC phenotypes [118]. Given these constraints, the rest of the review will maintain a particular focus on LSCs, presenting cell-intrinsic molecular aberrations and signaling circuitries that occur in the transcriptome, epi-transcriptome, epigenome, proteome, and metabolome of AML and CML LSCs.

### 3.2. AML

#### 3.2.1. Transcriptome and Transcription Factor Signaling of AML LSCs

AML LSCs are notoriously insensitive to standard chemotherapeutic treatments known to be effective against blasts [119]. The transcriptomic landscape accounting for AML LSC-mediated therapeutic resistance is dynamic and can be partially attributed to the capacity of LSCs to acquire distinct transcriptomic phenotypes. On the one hand, AML LSCs adopt a core “stemness” transcriptomic signature comparable to that of healthy HSCs, supporting the idea that AML LSCs may develop from gradual, stepwise transformations in healthy HSCs that eventually acquire the ability to give rise to hierarchically and clonally distinct leukemic cells [120]. Further resonating with this finding, Ng and colleagues generated a panel of 17 genes, including *GPR56* [121], *SOCS2* [122], and *CDK6* [123], that are specifically associated with LSC stemness, poorer overall survival, and worse initial treatment response [124]. On the other hand, the development of AML can be appreciated from a different light—rather than following a unidirectional, linear trajectory with LSCs representing the source of leukemogenesis, differentiated AML cells are also capable of reverting back to a relatively dedifferentiated, immature state via the suppression of the pioneer factor *PU.1*, for instance, suggesting transcriptional plasticity [125]. Nevertheless, such a dynamic transcriptomic signature can be responsible for LSC-mediated drug resistance. For example, in contrast to the transcriptomic signature of LSCs from patients with intermediate prognoses, CD34^+^CD123^+^CD3^−^CD19^−^ LSCs harvested from drug-resistant primary AML cells in patients harboring poor-risk prognosis with *TP53* alterations (TP53*Alt*) are enriched for anomalously activated STAT3 transcription factor signaling [126]. Indeed, the JAK/STAT signaling axis, specifically STAT3, promotes “stemness” in cancer stem cells and drives resistance to chemotherapy and radiotherapy, not only in AML, but also in other malignancies such as myxoid liposarcoma and colorectal cancer [127,128,129,130]. Other dysregulated transcriptomic networks underlying AML LSC-mediated therapy resistance and survival include the canonical nuclear factor κB (NFκB) pathway [131,132], depletion of the transcription factor *GLI3* in the hedgehog signaling pathway [133,134,135], and c-Myc dysregulation [136]. Of note, a single transcription factor mentioned above, such as c-Myc, can execute AML leukemogenesis in several ways through binding to distinct interaction partners, adding complexity to LSC biology. For instance, ERK- and MSK-mediated Sp1 and c-Myc interaction with the *survivin* promoter results in elevated expression of *survivin*, facilitating the acquisition of apoptotic resistance in AML LSCs [137]. Furthermore, c-Myc-induced upregulation of USP22 deubiquitinase promotes SIRT1 stability and subsequently confers protection against FLT3 inhibitors in FLT3-ITD AML LSCs [136]. Such transcriptomic abnormalities can also be coupled with molecular features that help LSCs and blasts to survive. For instance, elevated expression of calcitonin receptor-like receptor (*CALCRL*) is found to not only maintain AML LSC frequency, but to also facilitate cell-cycle progression in AML blasts [138]. These concerted molecular aberrations may not only lead to aggravated disease progression through blast expansion, but also the simultaneous persistence of AML LSCs, as evidenced in Table 2. 

#### 3.2.2. Epi-Transcriptome and Epigenome of AML LSCs

Epigenetic gene modulation has been a central theme in AML pathogenesis; however, its role in LSC survival and drug resistance remains vastly unknown. Contemporary efforts in identifying epigenetic regulators of LSCs have mainly centered on chromatin immunoprecipitation sequencing (ChIP-Seq) and large-scale functional knockdown screens. One such knockdown screen delineated the leukemogenic role of Enhancer of Polycomb genes *EPC1* and *EPC2*, as *EPC1* and *EPC2* knockdown deprived the MLL-fusion cell line THP-1 of clonogenic potential, likely through crosstalk with Myc [146]. Analysis of differentially methylated regions in engrafting AML LSCs and non-engrafting blast counterparts has revealed a predominantly hypomethylated epigenetic signature with marked enrichment of *HOXA* (*HOXA5*, *HOXA6*, *HOXA7*, *HOXA9*, and *HOXA10*) genes in LSCs [147]. In the specific case of MLL-rearranged AML, however, H3K4 methylation, besides being associated with MLL target genes such as *HOXA9* and *Meis1*, substantially correlates with the LSC maintenance transcriptional program gene set [148]. Furthermore, Yamazaki and colleagues also reported dynamic chromatin modifications, specifically H3K4me3, involvement with the transition from immature cancer stem cells to progenitor cells in AML [149]. Intriguingly, integrated genomic characterization of in vitro directed differentiation of AML-induced pluripotent stem cells (AML-iPSCs) revealed higher chromatin accessibility among intronic and intergenic regions within iLSCs (induced leukemic stem cells) compared to the iBlast (induced blast) fraction counterpart, hinting at the possibility of AML iLSCs requiring a greater degree of plasticity to maintain their unique molecular properties [150]. Indeed, such plasticity can be manifested as resistant AML LSCs resorting to enhancer switching to transcribe key survival genes [151]. 

Epigenetic regulation of mRNA is also critical to LSC survival [152]. For example, methyltransferase like 14 (METTL14), a key component of the N^6^-methyladenosine (m^6^A) complex that is responsible for methylating nitrogen-6 on the adenosine base, is indispensable for LSC self-renewal and frequency in vivo [153]. Similarly, YT512-B Homology N^6^-methyladenosine RNA binding protein 2 (YTHDF2), also known as an m^6^A reader, is uniquely required for AML LSC development and AML initiation and propagation, whereas its depletion has no impact on normal hematopoiesis [154]. Therefore, epigenetic reprogramming may represent yet another molecular sanctuary for the maintenance of LSCs, warranting increasing recognition of its clinical significance. 

#### 3.2.3. Proteome of AML LSCs

By taking into account post-translational modifications, cellular translational and protein degradation kinetics, proteomics overcomes the limitations of gene expression profiling by presenting accurate molecular snapshots of physical constituents of LSCs. However, proteomics studies on AML LSCs specifically are rather limited. Through a reverse phase protein array (RPPA), Kornblau and colleagues reported that CD34^+^CD38^−^ AML LSCs, in comparison to bulk and CD34^+^ cells, exhibited higher levels of P27, Mcl1, HIF1α, P53, Yap, and phospho-STATs 1, 5, and 6 [155]. Comparative tandem mass tag (TMT) multiplex proteomics analysis of CD34^+^CD123^+^ AML patient LSCs and control HSCs identified 171 significantly differentially expressed proteins. Subsequent enrichment pathway analysis linked these proteins to gene ontology clusters such as cellular response to endogenous stimulus and cellular response to metal ion, among many others [156]. The distinct methodologies adopted by the aforementioned studies make it difficult to consolidate their findings and draw definitive conclusions regarding the proteomic landscape of AML LSCs; obstacles include different surface markers used to identify LSCs as well as the uneven coverage and depth of protein identification and quantification based on the disparate techniques employed. Nevertheless, proteomics has been successful in unravelling metabolic vulnerabilities intrinsic to LSCs, as discussed below [157].

#### 3.2.4. Metabolome of AML LSCs

Numerous studies have reported metabolic alterations in the maintenance and drug-resistant nature of AML LSCs. Recently, Aasebø et al., using liquid chromatography-tandem mass spectrometry (LC-MS/MS), compared proteomic and phosphoproteomic profiles of AML cells derived from AML patients at the time of first diagnosis as well as at first relapse, and found significant enrichment of mitochondrial ribosomal subunit proteins, mitochondrial respiratory chain complex proteins, and proteins involved in mitochondrial metabolism, and higher phosphorylation level of nucleolar as well as nucleic-acid-binding metabolism proteins at first relapse [158]. This raises the possibility that drug-resistant, relapse-initiating LSCs may exploit mitochondrial dynamics to possibly reconcile survival and self-renewal with a predominantly quiescent state [159]. Specifically, AML LSCs are known to preferentially rely on oxidative phosphorylation (OXPHOS) as opposed to glycolysis to maintain cellular bioenergetics, and have managed to use BCL-2-depedent mechanisms to keep the production of reactive oxygen species (ROS), byproducts of OXPHOS, at bay, which is in alignment with their slowly proliferating, quiescent properties [160,161]. Indeed, several other papers have delineated the prominent antileukemic effects of the selective BCL-2 inhibitor Venetoclax in inhibiting OXPHOS in AML stem and progenitor cells [162,163]. Contrary to the conventional evidence, a study by Farge et al. revealed that CD34^+^CD38^−^ cells from BM of AML patient-derived xenograft (PDX) models neither increased nor exhibited a predominantly quiescent phenotype following AraC administration, and that the elevated OXPHOS machineries observed in chemo-resistant leukemic cells are specifically mediated by fatty acid oxidation [164]. Though the validity of the practice of defining LSCs solely based on limited surface antigens remains a topic of discussion, the phenomenon of fatty-acid-metabolism-driven OXPHOS is a critical theme in drug resistance. For instance, compared to drug-sensitive LSCs, LSCs obtained from Venetoclax- and Azacitidine-resistant specimens exhibited enhanced fatty acid oxidation and mitochondrial transport of fatty acids [165]. Further in line with this finding, Jones et al. reported that elevated fatty acid oxidation can compensate for and rescue amino-acid-deprivation-induced LSC killing in relapsed AML patients, further highlighting the critical dependence of AML LSCs on fatty acid metabolism to fuel OXPHOS [166]. Such significance of lipid homeostasis in AML LSCs is once again recapitulated as pharmacological inhibition of nicotinamide phosphoribosyltransferase (NAMPT), which depletes cellular nicotinamide adenine dinucleotide (NAD) and subsequently suppresses the conversion of saturated fatty acids to monounsaturated fatty acids, leads to the selective killing of LSCs over normal HSCs [167]. Collectively, metabolic diversion to OXPHOS and a preferential reliance on fatty acid metabolism may serve as integral molecular vulnerabilities of relapse-initiating AML LSCs, and that targeting specific pathways and metabolites within and upstream of OXPHOS may have the potential to safely and effectively eradicate LSCs. The aberrant multi-omics circuitry of AML LSCs is summarized in Figure 2a.

### 3.3. CML

#### 3.3.1. Transcriptome and Transcription Factor Signaling of CML LSCs

CML LSCs are intrinsically resistant to conventional TKI therapies such as imatinib (IM), nilotinib (NL), and dasatinib (DA). Single-cell analysis reveals two distinct subsets of BCR-ABL^+^ stem cells from patients who achieved hematological remission after being subjected to TKI treatments, with one group enriched for a quiescent gene signature and the other enriched for MYC and proliferation-associated gene sets, suggesting clonally segregated CML LSCs may account for differential sensitivities to TKIs, which can impact treatment outcomes [168]. Furthermore, transcriptomic profiling has revealed many signaling pathways integral for CML LSC survival, including the *TP53-cMYC* signaling network [169,170], arachidonate 5-lipoxygenase (*Alox5*) [171], tyrosine-protein kinase BLK [172], the NFκB pathway [173], the JAK2/STAT5 axis [174,175], the AHI-BCR-ABL-JAK2-DNM2 network [176,177], Wnt activation [178], the proinflammatory transforming growth factor (TGF)-β and tumor-necrosis factor (TNF)-α signaling pathways [179], and integrin-linked kinase signaling [180]. Of note, CML LSCs, in comparison to their normal counterparts, can generate alternative transcript isoforms for genes, particularly those involved in the cellular proliferation and p53 signaling pathways [181]. In addition to these cellular pathways, a broad range of microRNAs (miRNAs) are known to regulate transcriptomic dynamics to confer drug resistance in CML LSCs. Indeed, miRNome analysis of LSC-enriched CD34^+^CD38^−^CD26^+^ and normal HSCs obtained from chronic-phase CML patients seemed to show decreased levels of total miRNAs compared to that of primitive cells from healthy donors [182]. CD34^+^ cells from IM non-responder patients have significantly lower expression of *miR-185*, the deficiency of which upregulates PAK6 and mediates TKI resistance [183]. Furthermore, miR-21 can interact with the PI3K/AKT pathway to confer therapeutic resistance to IM in CD34^+^ stem and progenitor CML cells [184]. Furthermore, extracellular vesicles derived from endothelial cells can also mediate intercellular transport of miR-126 to CML LSCs, further driving LSC persistence and quiescence [105]. In summary, CML LSCs can utilize both cell-intrinsic (aberrant miRNA and transcription factor signaling dynamics) and cell-extrinsic mechanisms (crosstalk with BM endothelial cells) to support their survival, persistence, and drug resistance, showcasing their transcriptional plasticity.

#### 3.3.2. Epi-Transcriptome and Epigenome of CML LSCs

CML LSCs can manipulate the epigenome to enable TKI resistance. DNA methylation has been a topic of investigation in CML LSCs. Polycomb repressive complexes (PRC) are critical for the maintenance of healthy HSCs by suppressing the transcription of genes involved in cell proliferation and differentiation via histone tri-methylation and mono-ubiquitination [185]. Dysregulation and mutations of PRC subunits results in various hematological malignancies, including T-cell ALL, AML, and blast-crisis CML [186,187,188]. For instance, *EZH2*, the catalytic subunit of PRC2, is overexpressed in CML leukemia-initiating cells (LICs), and its depletion in LICs results in decreased capacity to form secondary leukemia upon transplantation in a murine model [189]. In line with this evidence, inhibition of EZH2 sensitizes CML LSCs to TKIs [190]. Furthermore, the methylome of CML patients in the chronic phase and blast crisis reveals more than a 10-fold increase in differentially methylated CpG sites, with prominent downregulation of tumor suppressors. Additionally, global DNA methylation profiles of CML remission samples are similar to those from healthy donors as opposed to those obtained at the time of diagnosis [191]. These pieces of evidence collectively suggest that aberrant methylation landscape, specifically hypermethylation, can serve as a unique epigenetic hallmark of CML disease progression and may functionally implicate CML LSCs [192]. Indeed, the survival, engraftment capability, self-renewal, and low TKI sensitivity intrinsic to CML LSCs has been partially attributable to various methyltransferases, such as DNA methyltransferase 1 (DNMT1) [193], protein arginine methyltransferase 5 (PRMT5) [194], and protein lysine methyltransferase G9A [195], establishing the rationale for clinical implementation of CML-based epigenetic therapeutics. 

#### 3.3.3. Proteome of CML LSCs

Recent proteomic interrogation of CML LSCs has largely focused on the identification of dysregulated kinase activities in LSCs as numerous BCR-ABL-independent mechanisms of resistance in CML begin to emerge [196,197]. For instance, quantitative phosphoproteomic analysis of IM-resistant K562 identified Tpl2 (tumor progression locus)-derived phosphopeptide as being highly dysregulated compared to IM-sensitive counterparts, and found a similar scenario in CD34^+^ CML patient BM cells, identifying MEK-ERK, Src family kinases, and NFκB signaling as potential mediators of IM resistance [198]. Of note, BCR-ABL may also likely establish molecular crosstalk with other kinases such as JAK2 [199] and JAK2/STAT5 [174] to maintain CML LSC properties. Exemplifying this notion even further is the discovery that AHI-1, a scaffolding protein found to be highly deregulated in CML LSCs, directly interacts with multiple proteins, including BCR-ABL, JAK2, and DNM2, to mediate CML LSC properties and confer TKI resistance [176,177,200]. Given the well-documented pro-survival role of Src and STAT phosphorylation in CML [201], it is interesting, however, that phospho-kinase proteomic profiling of CD34^+^ cells from chronic-phase CML patients by Ricciardi et al. reported that, in comparison to normal CD34^+^ cells, leukemic CD34^+^ progenitor cells exhibited significantly decreased phosphorylation of several Src (Lyn, Lck, and Fyn) and STAT (STAT 2, 5a, 5b, and 6) family kinases and increased phosphorylation of p53 [202]. However, it should also be noted that these aforementioned proteomics studies were, once again, conducted on stem and progenitor cells with different sets of surface antigens and from different sources (primary vs. established cell lines), offering room for reconciliation of seemingly contrasting data.

#### 3.3.4. Metabolome of CML LSCs

Proteomic comparison of CD34^+^CD38^−^CD26^+^ and CD34^+^CD38^−^CD26^−^ cells identified increased levels and enrichment of proteins responsible for macromolecule metabolism (PPARD, M3K14, HNF6, and LPH) in CD34^+^CD38^−^CD26^+^ cells, which represent the CML LSC fraction [203]. Metabolic analysis of stem-cell enriched CD34^+^ and CD34^+^CD38^−^ and differentiated CD34^−^ cells from CML patients has also indicated that CML LSCs require upregulated oxidative metabolism for their survival [204]. Nonetheless, a recent study challenged this notion, claiming that OXPHOS was also upregulated, as opposed to being downregulated, in metformin-mediated killing of CML stem and progenitor cells, suggesting the dispensability of OXPHOS in CML stem and progenitor cell survival [205]. Of note for this particular study, the potentially pleiotropic and uncharacterized effects of metformin may play a role in yielding such an interpretation. Nevertheless, it is clear that metabolic reprogramming of drug-resistant CML LSCs can occur through both epigenetic and genetic mechanisms. For instance, SIRT1 de-acetylase depletion leads to reduced mitochondrial respiration in CML stem/progenitor cells, likely involving the SIRT1 substrate PGC-1α [206]. Furthermore, a strategic combination of transcriptomic analysis, ChIP-seq, and pathway analysis reveals that STAT3 can transcriptionally mediate metabolic genes involved in glycolysis, carbohydrate metabolism, and the pentose phosphate pathway in TKI-persistent LSCs [201]. Unlike low-ROS AML LSCs, however, CML LSCs appear to depend on elevated ROS to foster genomic instability, which leads to TKI-resistant BCR-ABL1 mutations (E255K, T315I, and H396P) [207,208,209,210]. The aberrant multi-omics circuitry of CML LSCs is summarized in Figure 2b.

## 4. Potential Therapeutic Strategies to Combat LSCs

The abundance of molecular and phenotypical aberrations associated with LSCs offers a wealth of promising therapeutic targets. Current therapeutic designs have focused on drugging surface biomarkers selectively overexpressed on LSCs, antagonizing the protective BM microenvironment niche to dismantle LSC dormancy, blocking signal transduction to re-sensitize resistant LSCs to available chemotherapeutics, and even expediting the drug supply pipeline through drug repurposing. Evidently, growing insight into the biological properties and prognostic values of LSCs have prompted the implementation of many clinical trials and have laid critical groundwork for the development of more effective, personalized, scalable, and less-toxic therapeutic strategies.

### 4.1. Surface Antigen-Based Immunotherapies

Interrogation of LSC surface antigen milieu has encouraged the development of distinct functional categories of antibodies, namely mono-specific and bi-specific antibodies, as well as antibody-drug conjugates (ADCs). Mono-specific antibodies against prominent LSC markers are relatively uncommon. This can presumably be due to several reasons, including but not limited to the fact that there is likely no singular “panacea” LSC surface marker whose targeting will disarm all LSC phenotypical and functional characteristics and exert broad-spectrum antileukemic effects for all patients. To resolve this dilemma, current therapeutic efforts center on drugging multiple LSC surface markers at once in the form of bi-specific antibodies and ADCs. Nevertheless, a few clinically employed mono-specific antibodies that are also known to target LSC surface antigens include ^213^Bi-lintuzumab (anti-CD33) [211], Talacotuzumab (anti-CD123) [212], Magrolimab (anti-CD47) [213,214], and daclizumab (anti-CD25) [215,216], with some of them demonstrating exceptional anti-leukemic effects against residual and resistant hematological cancer cells. However, it is crucial to acknowledge that the identification and practicality of targeting LSC surface antigens, at least via mono-specific antibodies, remains largely empirical. Bi-specific antibodies, on the other hand, possess greater flexibility in accommodating substrates from either the same target or distinct targets. Flotetuzumab, a bi-specific antibody that recognizes CD123 and CD3, in addition to having the potential to target leukemic blasts and LSCs overexpressing CD123, can also utilize CD3 to activate T cells and redirect their cytotoxicity towards CD123^+^ AML cells, rendering it a salvage immunotherapy for refractory AML patients [217,218]. A few other bi-specific antibodies, including AMG330 (anti-CD33 and anti-CD3) [219] and blinatumomab (anti-CD19 and anti-CD3) [220], have also been uncovered to elicit T-cell mediated immunity to exert sustained anti-leukemic effects. Furthermore, as another therapeutic modality, ADCs are typically composed of a cytotoxic payload chemically connected to a monoclonal antibody via a biodegradable linker, whereby the antibody component mediates the specific delivery of the payload into target cells to minimize unintentional cytotoxicity. A classic example is gemtuzumab ozogamicin (GO) [221,222], which consists of calicheamicin linked to an anti-CD33 antibody and has entered multiple phase III clinical trials for adult and pediatric AML patients harboring diverse cytogenetic phenotypes, such as *nucleophosmin1* (*NPM1*)-mutated [223] and *lysine methyltransferase 2A* (*KMT2A*)-rearranged AML subtypes [224,225]. Importantly, it has been shown that GO is effective in managing minimal residual disease and drastically reduces chemo-residual leukemic-initiating cells upon incorporation into conventional induction chemotherapy [226,227]. Recently, SGN-CD33A, a humanized ADC composed of the DNA cross-linking agent pyrrolobenzodiazepine dimer and anti-CD33, was developed with more enhanced therapeutic efficacy than that of GO, especially for AML subtypes associated with poor prognosis and entailing a multi-drug resistant phenotype [228]. Another example of AML LSC- and leukemic blast-specific ADC is CLT030, whereby a DNA-binding payload is covalently linked to anti-CLL1, creating a site-specific avenue for drug delivery. As a result, the administration of CLT030 leads to decreased LSC colony formation and even presents a more favorable toxicity profile than that of CD33-ADC sharing the identical payload as CCL1-ADC [229].

### 4.2. Small-Molecule Inhibitors

As discussed previously, LSC-driven hematological malignancies present multi-dimensional molecular abnormalities in their transcriptome, epigenome, proteome, and metabolome. Many of these alterations can then be harnessed as the basis for therapeutic design of small-molecule inhibitors. For instance, recently, Jiang and colleagues reported the potent anti-leukemic effects of the highly selective AXL kinase inhibitor SLC-391 on MLL-fusion AML stem and progenitor cells in vitro and in vivo [230]. Additionally, jumonji domain modulator #7 (JDM-7) binds and inhibits histone lysine demethylase JMJD1C and effectively downregulates LSC self-renewal gene HOXA9 to selectively decrease colony formation of leukemic cells in vitro in MLL-rearranged AML [231]. Intriguingly, small-molecule inhibitors are also designed to disarm leukemic niche signaling. For example, Dynole 34-2, which inhibits Dynamin GTPase activity, blocks receptor-mediated endocytosis critical for niche-mediated growth-factor signaling in pre-LSCs [232]. Furthermore, targeting the epigenetic m^6^A modification in AML by inhibiting the catalytic activity of the METTL3 methyltransferase by the small-molecule inhibitor STM2457 impairs engraftment potential in murine models of AML [233]. In IM-resistant CML, pharmacological inhibition of ubiquitin-specific peptidase 47 (USP47) with P22077 reduces the percentage of CD34^+^CD38^−^ cells in secondary BM transplantation and inhibits colony-forming activity of CD34^+^ cells from IM-resistant CML patients while sparing normal CD34^+^ cells [234]. To meet the burning clinical need of eradicating LSCs, research on drug repurposing is also rapidly expanding. For example, proscillaridin A, predominantly indicated for heart failure, has been empirically demonstrated to kill MYC-overexpressing LSCs in both T-ALL and AML models, potentially through downregulating acetylation of MYC target genes [235]. Another interesting case of drug repurposing is that of salinomycin. As an antimicrobial drug, salinomycin is found to confer cytotoxicity against a broad spectrum of cancer stem cells [236]. In MLL-rearranged AML, sub-micromolar treatment of salinomycin on human and mouse primary leukemia cells led to reduced colony formation while sparing normal samples, indicating anti-LSC activity of salinomycin [237]. Intriguingly, powerful in silico analysis of AML LSC gene expression signatures crossed with drug–gene interaction datasets has yielded diverse cohorts of repositionable drugs, potentially increasing the diversity and accessibility of a drug repertoire that inhibits LSCs to better manage residual malignancies [238]. 

### 4.3. Combination Therapies

Combination therapies represent a major research hotspot. The recognition that cancer cells are often more susceptible to disruption of multiple pathways at once and the possibility of combining several targets to more precisely capture a specific oncogenic signature to minimize off-target effects has led to the advent of numerous strategic combination regimens. Among several recently developed combination therapies (Table 3), oxidative metabolism-based therapeutics have garnered astounding popularity due to the shared dependence of AML and CML LSCs on OXPHOS for survival. To highlight a few discoveries, in AML, the BCL-2 inhibitor Venetoclax synergizes with ribonucleoside analog 8-chloro-adenosine (8-Cl-Ado) to decrease OXPHOS of CD34^+^CD38^−^ LSC-enriched cells [163]. Furthermore, Venetoclax in combination with the hypomethylating agent Azacytidine results in decreased electron transport chain complex II activity, suppressing OXPHOS and consequently leading to the death of AML LSCs [143]. A recurring theme herein is the incorporation of the BCL-2 inhibitor Venetoclax into combination regimens against AML LSCs. Indeed, clinical evaluation of AML patient LSC profiles has indicated that elderly AML patients may particularly benefit from Venetoclax combination therapy [239]. Apart from AML, in CML, combined blockade of BCL-2 by Venetoclax and BCR-ABL tyrosine kinase using a TKI effectively eradicates CML LSCs in vitro and in vivo [240]. Interestingly, combination therapy consisting of the integrin-linked kinase (ILK) inhibitor QLT0267 and DA effectively inhibited the growth of primitive CML cells by downregulating oxidative metabolism and mitochondrial dynamics while sensitizing refractory patient LSCs to TKI therapy in vitro and in a PDX model [180]. In line with this, tigecycline, a mitochondrial protein translation inhibitor, in conjunction with IM, selectively eradicates CML LSCs both in vitro and in vivo [204]. Furthermore, using an advanced drug/proliferation screen, Lai et al. uncovered a pro-survival role for protein phosphatase 2A (PP2A) in TKI-nonresponder cells, and that the inhibition of PP2A impaired survival of these cells and sensitized them to TKIs, inducing a dramatic loss of several key proteins, particularly β-catenin [241]. Remarkably, the clinically validated PP2A inhibitors LB100 and LB102, in combination with TKIs, act synergistically to inhibit the growth of CML LSCs [241] and BCR-ABL^+^ ALL patient cells [242].

### 4.4. Theoretical Nanomedicines

Recent years have witnessed the development of nanoparticles as novel drug delivery systems against LSCs. A variety of nanocarriers have been engineered to accommodate a plethora of drug molecules and are able to deliver them to targets with superior precision [254]. For instance, curcumin, an anti-leukemic compound with low water solubility, can be loaded into poly (lactic-co-glycolic acid) PLGA/poloxamer nanoparticles and conjugated to anti-CD123, a selective LSC surface antigen, to reliably target AML LSCs in vitro [255]. Refining target specificity in the context of nanocarriers can also be achieved by endowing the nanoparticle with a tissue-targeting binding peptide. Introduction of a tartrate-resistant acid phosphatase (TRAP)-binding peptide to amphiphilic poly (styrene-*alt*-maleic anhydride)-*b*-poly(styrene) (PSMA-*b*-PS)-based nanoparticles loaded with Micheliolide analog 64 drives precise delivery of drug-loaded nanoparticles to BM and leads to sustained reduction of marrow LSCs in a murine blast crisis CML model [256]. In addition to encapsulating cytotoxic agents, nanoparticles may also deliver therapeutic siRNAs to target cells, functioning as critical agents of modern gene therapy [257,258]. Nevertheless, elucidating novel nano-constructs and conducting robust clinical testing are still needed to further broaden the scope and applicability of LSC-targeting nanomedicines. 

## 5. Conclusions

From theoretical postulation of the existence of LSCs to successful isolation and subsequent molecular characterization of LSC-enriched populations in hematological malignancies, researchers have made considerable strides towards understanding the molecular behaviors that contribute to the self-renewal, quiescence, and drug-resistant properties of LSCs. Even though LSCs represent only a minor fraction of the total leukemic bulk cell population, LSCs are capable of derivatizing unique surface antigen milieu to coordinate intracellular signaling, mediating crosstalk and physical interactions with the stromal microenvironment, and rewiring transcriptomic, epigenomic, proteomic, and metabolomic profiles to facilitate leukemogenesis. Owing to rapidly developing multi-omics and single-cell technologies, researchers can now harness integrated conceptual frameworks of the molecular aberrations intrinsic to LSCs to devise meaningful therapeutics, generating a multitude of antibodies, small-molecule-based inhibitors, combination therapies that possibly benefit patients who cannot tolerate conventional chemotherapeutics, and innovative drug delivery systems such as nanocarriers. These transformative therapeutic approaches reflect the multi-layered complexity characteristic of LSCs, underscoring the necessity of expediting the drug discovery and supply pipeline to meet the clinical demands of patients living with hematological malignancies.

## Figures and Tables

**Figure 1 biomedicines-10-01841-f001:**
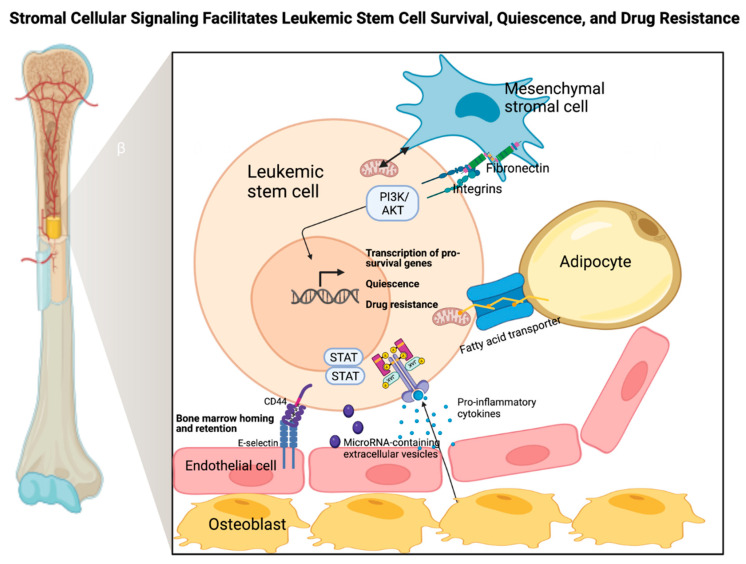
**Stromal cellular signaling facilitates leukemic stem cell (LSC) survival, quiescence, and drug resistance.** LSCs engage in bidirectional crosstalk with multiple BM cellular constituents. E-selectin expressed on the surface of endothelial cells interacts with CD44 on LSCs to drive LSC homing and retention in the protective BM microenvironment, sheltering LSCs from therapeutic insults. Furthermore, endothelial cells also release microRNA (miRNA)-containing extracellular vesicles to further enrich the quiescence phenotype of LSCs. Osteoblasts primarily release proinflammatory cytokines that lead to the transcription of genes implicated in LSC survival, self-renewal, and quiescence. Mesenchymal stromal cells are known to physically transfer mitochondria to LSCs via nanotubes to repair and replace damaged mitochondria with new ones inside LSCs, potentially helping LSCs evade apoptosis. Mesenchymal stromal cells can also activate pro-survival integrin-mediated signaling in LSCs, involving the PI3K/AKT pathway. Moreover, adipocytes assist in the rewiring of LSC metabolism, supplying free fatty acids to fuel oxidative phosphorylation, a known metabolic dependency of LSCs.

**Figure 2 biomedicines-10-01841-f002:**
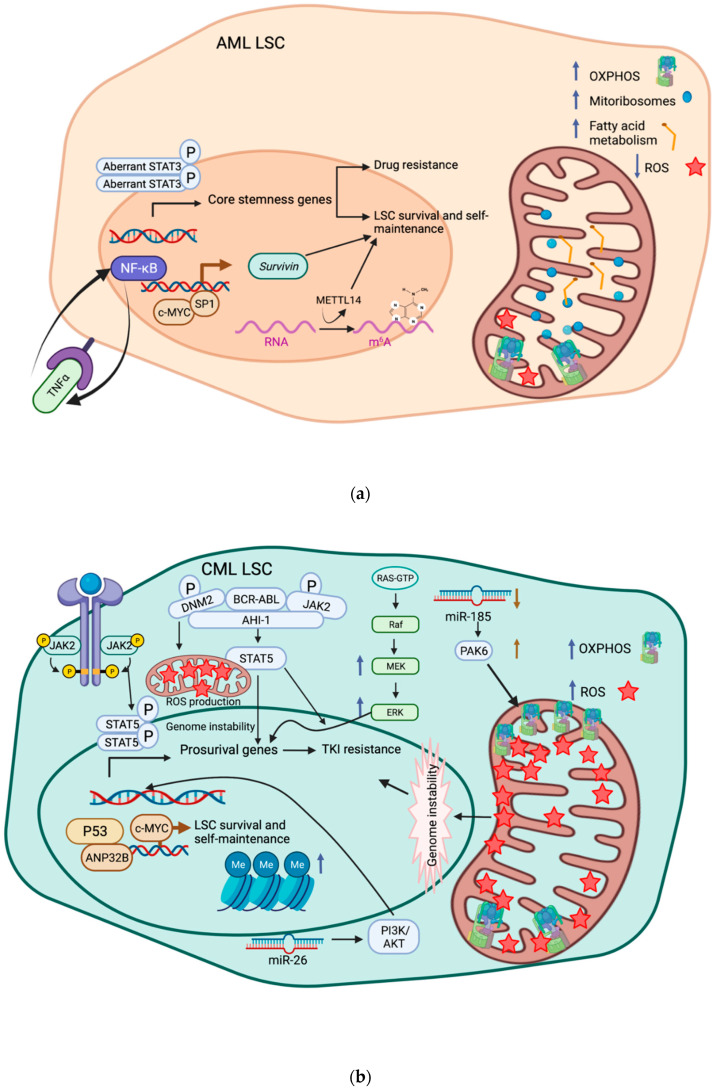
**Multi-omics circuitry of AML and CML LSC-mediated drug resistance.** (**a**) Notable transcriptomic features of AML LSCs include dysregulated transcription factors, such as STAT3, the aberrant activation of which is associated with the transcription of core stemness genes; constitutive NFκB activation, which can be mediated by a self-sustaining, autocrine positive feedback loop with tumor necrosis factor-α (TNF-α); and aberrant c-Myc activity, which, along with sp1, enhances transcription of *survivin*, concertedly driving LSC survival, self-maintenance, and drug resistance. Epigenetically, m^6^A RNA modification by METTL14 is essential for LSC self-renewal and frequency in vivo. In regard to the proteomic and metabolomic landscapes of AML LSCs, AML LSCs tend to harbor high abundance of mitochondrial ribosomes, also known as mito-ribosomes, to facilitate translation of mitochondrial and oxidative phosphorylation (OXPHOS) machineries, to which fatty acid oxidation contributes a great deal. Interestingly, AML LSCs generally maintain modest to low levels of reactive oxygen species (ROS), in alignment with their generally quiescent state. (**b**) CML LSCs have also shown oncogenic aberrations to transcription factor activities, including JAK2/STAT5 signaling, which activates transcription of pro-survival genes and confers TKI resistance. Dysregulation of P53 signaling by c-Myc or Acidic Nuclear Phosphoprotein 32 Family Member B (ANP32B) fosters LSC survival and self-maintenance. Furthermore, the AHI-1-BCR-ABL-JAK2-DNM2 signaling network facilitates multiple features of CML LSC survival and drug resistance, such as activating STAT5 signaling, increasing ROS production, and promoting genome instability, all of which drive overall LSC proliferation and resistance to therapy. The abilities of CML LSCs to self-renew and to resist against TKI therapy can further be enhanced by epigenetic mechanisms such as increased global DNA methylation and dysregulated miRNA milieu (e.g., downregulation of miR-185 or increased level of miR-26). Particularly, downregulation of miR-185 increases its target PAK6 level, which leads to increased OXPHOS capacity and ROS production of CML LSCs. Aberrant kinase activation, such as ERK/MEK, may partially account for proteomic anomalies underlying LSC survival. Like AML LSCs, CML LSCs tend to rely on OXPHOS to maintain cellular bioenergetics. However, unlike AML LSCs, CML LSCs thrive under elevated ROS, as it triggers further genome instability and potentially gives rise to TKI-resistant BCR-ABL mutations such as T315I.

**Table 1 biomedicines-10-01841-t001:** Surface Antigen Phenotypes of AML and CML LSCs.

Malignancy	Surface Antigen Phenotype	Significance	References
AML	CD34^+^CD38^−^	Denotes a primitive subpopulation of stem/progenitor cells in AML	[6]
	CD33^+^	Selectively overexpressed in AML patients compared to healthy HSCs	[21]
	CD123^+^	Selectively overexpressed on AML cells and potentially facilitates STAT5 activation	[22,23,24,25]
	CD47^+^	Assists AML LSCs with apoptotic evasion via phagocytic inhibition of circulating macrophages and dendritic cells	[26,27,28]
	CD96^+^	Upregulated in AML cells and enriches LSC activity	[29]
	CD99^+^	Selectively overexpressed on AML LSCs, particularly at disease relapse	[30]
	CD45^dim^CD34^+^CD38^−^CD133^+^	Enriched in AML BM samples and associated with poor overall and event-free survival of AML patients	[31]
CML	CD34^+^CD38^−^	Denotes a primitive subpopulation of stem/progenitor cells in CML	[36,38]
	CD33^+^	Chronic phase CML patients exhibit a roughly 10-fold higher expression of CD33 compared to CD34^+^CD38^−^ cells from healthy individuals	[42]
	IL1RAP	Upregulated in CD34^+^ and CD34^+^CD38^−^ CML cells. Further upregulated in accelerated and blast crisis phases compared to chronic phase	[43]
	CD34^+^CD38^−^CD26^+^	Exhibits repopulating capacity in NSG mice and upregulated in imatinib-nonresponders	[44,45]
	Lin-CD34^+^CD38^−/low^CD45RA^−^cKIT^−^CD26^+^	Denotes CML LSCs that are particularly insensitive to TKI therapies	[46]

**Table 2 biomedicines-10-01841-t002:** Exemplary LSC Resistance Mechanisms to Clinically Approved AML Therapies.

Drug Class	Name	Potential AML LSC Resistance Mechanisms	References
Anthracyclines	Doxorubicin	NCAM1-mediated constitutive activation of the pro-survival MAPK signaling pathway	[139]
	Daunorubicin	Potentially via upregulated multi-drug resistance transporters	[140,141]
	Idarubicin	CALCRL-mediated DNA damage repair and cell-cycle progression	[138]
Hypomethylating Agent	Azacitidine	Potential deposition of LSCs in the protective BM microenvironment; enhanced OXPHOS machineries	[142,143]
IDH Inhibitor	Ivosidenib/enasidenib	Expression of stemness-associated gene signatures	[57]
FLT3 Inhibitor	Sorafenib	Enhanced stromal interaction and diminished activation of pro-survival signaling mediated by the SDF-1α-CXCR4 axis	[144,145]

**Table 3 biomedicines-10-01841-t003:** Combination therapeutic approaches against LSC-enriched cells in AML and CML.

Malignancy	Combination Regimen	Mechanism of Action	Type of LSC-Enriched Population Targeted	Reference
AML	Venetoclax + 8-chloro-adenosine	Decreased fatty acid oxidation and OXPHOS	CD34^+^CD38^−^ primary AML blasts	[163]
	Venetoclax + Azacitidine	Decreased electron transport chain complex II function and OXPHOS	CD34^+^CD38^−^Lin^−^CD123^+^ AML LSCs	[143]
	Venetoclax + SLC-391	Perturbation of OXPHOS	CD34^+^ AML stem and progenitor cells	[230]
	Venetoclax + GDC-0980 (PI3K/mTOR inhibitor)	Inactivation of AKT/mTOR/p70S6K and induction of intrinsic apoptosis	CD34^+^CD38^−^CD123^+^ AML stem and progenitor cells	[243]
	Venetoclax + CT7001 (CDK7 inhibitor)	LSC-targeting mechanism likely involves the disruption of dynamic coordination of GPR56 with Wnt, hedgehog, and epithelial-mesenchymal transition signaling network	Sustained suppression of PDX human CD34^+^GPR56^+^ AML cells isolated from NSG murine BM	[244]
	AT-101 (BCL-2 inhibitor) + idarubicin	Inhibition of DNA damage repair	CD34^+^CD38^−^ KG1α and Kasumi-1 cell lines; CD34^+^ primary cells	[245]
	Tenovin-6 + quizartinib (AC220)	Inhibition of SIRT1-mediated downregulation of p53	FLT-ITD^+^ CD34^+^ AML progenitors	[136]
	Chidamide + apatinib	Reduction of mitochondrial oxidative metabolism	CD34^+^CD38^−^ KG1α cells;CD34^+^ primary AML cells	[246]
	BAY1436032 (mutant IDH1 inhibitor) + azacitidine	Decreased MAP kinase and retinoblastoma/E2F signaling and downregulation of 11 genes from LSC17 gene panel	AML leukemic stem cells characterized by serial limiting dilution transplantation	[247]
CML	DA + bosutinib	Synergistic apoptotic induction and blockage of LYN, KIT, and PDGFRα kinase signaling	Patient-derived CD34^+^CD38^−^ CML stem cells	[248]
	QLT0267 + DA	Downregulation of OXPHOS to sensitize primitive TKI-resistant CML cells	Refractory, quiescent CD34^+^ and CD34^+^CD38^−^ CML patient LSCs	[180]
	Plasminogen activator inhibitor-1 (PAI-1) TM5614 + imatinib	Displacement of CML LSCs from the protective BM microenvironment	Lin^−^c-kit^+^Sca-1^+^ CML LSC cells	[249]
	Lys05/PIK-III (autophagy inhibitors) + NL	Loss of quiescence of CML stem cells	CD34^+^CD133^+^ primary CML cells	[250]
	MRT403 (ULK1/2 inhibitor) + IM	Loss of quiescence and increase of ROS by inducing metabolic shift from glycolysis to oxidative metabolism	CD34^+^ primary CML cells	[251]
	Tenovin-6 + IM	Increase in p53 acetylation and p53-mediated transcriptional activity	CD34^+^CD38^−^ and CD34^+^CD38^+^ stem and primitive CML progenitor cells	[252]
	MAKV-8 (HDAC inhibitor) + IM	Reduction of c-MYC expression; decreased BCR-ABL and STAT5 phosphorylation	CD34^+^CD38^−^ primary CML cells	[253]
	Venetoclax + NL	Cooperative inhibition of BCL-2 and BCL-X_L_/MCL-1 by nilotib and Venetoclax	CML bulk, CD34^+^CD38^−^, CD34^+^CD38^+^, and quiescent CD34^+^ blast crisis patient cells	[240]
	LB100/LB102 + IM/DA	Disruption of AHI-1-mediated signaling, particularly β-catenin	CD34^+^ CML stem and progenitor patient cells	[241]

## Data Availability

Not Applicable.

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
