# Peer review of "Properties of Leukemic Stem Cells in Regulating Drug Resistance in Acute and Chronic Myeloid Leukemias"

_biomedicines, 2022, doi:10.3390/biomedicines10081841_

Round 1
Reviewer 1 Report
The Review article by Zhai& Jiang on the “Properties of Leukemic Stem Cells in Regulating Drug Resistance in Human Leukemia” is generally well written and an enjoyable read. The authors found a nice pitch from historical development of the stem cell field to current insights into LSC biology. Yet some sections are sub-structured where sub-structure may be a hindrance to fluency. In addition, it remains unclear to the reviewer why CML has several own sections, which is also not reflected by the title. Overall, the review –with some restructuring and shortening- will be of interest for the readership with summarizing a very intense field.
Comments:
Preferentially titles stay short but this review focusses on myeloid leukemia (AML/CML). One could replace “Human” for “Myeloid” since –from this reviewers point of view- stating a focus on human is unnecessary and some studies refer to murine approaches.
Figures should be moved to the beginning of the respective sections to provide a first visual overview and guidance.
Sections on the niche and signaling would benefit from some shortening, which may already happen when restructuring and not taking CML as a separate topic.
Intro mentions the early work by Dick and colleagues and it may be noted that it later turned out to have an antibody-induced bias
Pre-leukemic stem cells are a major topic today and have been shown to pose the risk for relapse. Depending on the point of view, it may even be hard to discern them from LSCs. It is mandatory to have a brief section on these cells.
Section on surface antigens would benefit from slight shortening and a summarizing table or figure. The information provided in the specific section on surface antigens in CML does not warrant an own section but could be combined
The niche is a highly complex and controversial topic. However, from this reviewers’s point of view an integrated evaluation makes most sense. In contrast, the manuscript splits different cell types into individual sections and thereby somewhat ignores interactions.
Moreover, the niche is supposedly dynamic and was both shown to be remodeled in some instances of leukemia while being even an initiator of transformation in other cases (e.g. Wnt-overactivation, Dicer1 defects). Both aspects have to be covered by the review.
While being well structured in general, the section on Cell intrinsic signaling starts with a section of doubtful structure. It is recommended to first approach the different ways how LSCs may be generated (from HSC or by dedifferentiation) and then go into the concrete details. And while it is of course hard to be discrete on such topics, this first section jumps between signaling and transcriptomics. This also applies to epigenetic changes, which are just another way to regulate the transcriptome and subsequent events and here the authors include epi-transcriptomics. It is worth to consider a more general section on LSC induction and then move “naturally” from epigenomics to transcriptomics, epi-transcriptomics, proteomocs and last the metabolism.
Why is CML treated separately? With the current title and topic, CML should be integrated.
Consider to re-arrange therapies to their respective topics. E.g. surface antigen directed therapies could follow the section on immune-phenotype. To this reviewer it makes more sense to present the therapies with the biology instead of taking all therapies in sum as an individual section. Please also note that the section of small molecule inihibitors is rather limited, yet it would need an individual review to cover all potential options presented today and should not be extended but also included where the biology is presented (which is already the case with some strategies like Bcl2-inhibition). The same applies for combinatorial therapies.
Author Response
Reviewer 1:
Comments and Suggestions for Authors
The Review article by Zhai& Jiang on the “Properties of Leukemic Stem Cells in Regulating Drug Resistance in
Human Leukemia” is generally well written and an enjoyable read. The authors found a nice pitch from historical
development of the stem cell field to current insights into LSC biology. Yet some sections are sub-structured
where sub-structure may be a hindrance to fluency. In addition, it remains unclear to the reviewer why CML has
several own sections, which is also not reflected by the title. Overall, the review –with some restructuring and
shortening- will be of interest for the readership with summarizing a very intense field.
We thank the reviewer for valuable comments and systematic evaluation of the review. We also appreciate and
acknowledge the reviewer’s concern that certain sub-sections of the review may seem incongruent with the
overarching title of the manuscript. To reconcile these incongruencies and more accurately reflect the scope of
the review, we have modified the title of the manuscript to “Properties of Leukemic Stem Cells in Regulating
Drug Resistance in Acute and Chronic Myeloid Leukemias.” This way, we hope to inform the broader readership
of Biomedicines that this review will primarily highlight phenotypical, cellular, and molecular networks of
leukemic stem cells in the context of acute and chronic myeloid leukemias, hence allowing AML and CML to be
presented as separate pathological conditions that are in alignment with current empirical, clinical, and
diagnostic practices and standards.
Comments:
1) Preferentially titles stay short but this review focusses on myeloid leukemia (AML/CML). One could replace
“Human” for “Myeloid” since –from this reviewers point of view- stating a focus on human is unnecessary and
some studies refer to murine approaches.
We thank the reviewer for this valuable feedback. Indeed, this review has been largely focusing on stem cell
biology underlying acute and chronic myeloid leukemias; therefore, to reflect this focus, the title of the review
has now been modified, as per reviewer’s suggestion, to “Properties of Leukemic Stem Cells in Regulating Drug
Resistance in Acute and Chronic Myeloid Leukemias”.
2) Figures should be moved to the beginning of the respective sections to provide a first visual overview and
guidance.
We thank the reviewer for this insightful perspective. To provide a first visual overview of the sections, we have
moved Figure 1 to the beginning of section 2, and Figures 2a and 2b to the beginning of section 3.
3) Sections on the niche and signaling would benefit from some shortening, which may already happen when
restructuring and not taking CML as a separate topic.
We thank the reviewer for this valuable suggestion. Since the title of the review has been modified, as per
reviewer’s comment, to reflect the extensive focus on AML and CML LSCs, sections on the niche and signaling,
B C C A N C E R R E S E A R C H
675 West 10th Avenue Tel: 604.675.8000
Vancouver, BC, Canada V5Z 1L3 Fax: 604.877.0712
www.bccancer.bc.ca
upon incorporation of other reviewer’s insightful suggestions, can now more appropriately serve the central
theme and title of the review manuscript.
4) Intro mentions the early work by Dick and colleagues and it may be noted that it later turned out to have an
antibody-induced bias
We thank the reviewer’s thoughtful comment. Indeed, early studies are not immune to potential pitfalls and
biases resulting from techniques used and assay conditions. These potential biases are now discussed in page 3
(lines 84-88).
5) Pre-leukemic stem cells are a major topic today and have been shown to pose the risk for relapse. Depending
on the point of view, it may even be hard to discern them from LSCs. It is mandatory to have a brief section on
these cells.
We would like to thank the reviewer for this valuable comment. We have now added a brief section (Section 3.1
page 12, from line 476 to line 501) on pre-leukemic stem cells (pre-LSCs) and outlined several constraints and
challenges associated with studying these intriguing cells in myeloid malignancies, including phenotypical
distinction of pre-LSCs from LSC and HSC counterparts, as the reviewer graciously pinpointed.
6) Section on surface antigens would benefit from slight shortening and a summarizing table or figure. The
information provided in the specific section on surface antigens in CML does not warrant an own section but
could be combined
We thank the reviewer’s thoughtful comment. A brief summarizing table (Table 1) has been added to the section
on surface antigens to provide readers with a quick visual outlook of the text.
7) The niche is a highly complex and controversial topic. However, from this reviewers’s point of view an
integrated evaluation makes most sense. In contrast, the manuscript splits different cell types into individual
sections and thereby somewhat ignores interactions.
We thank the reviewer’s critical input. Indeed, the layout of the section may potentially mislead readers into
thinking that LSC stromal interactions are initiated from a single, individual cell type. Rather, our intention has
always been that multiple cell types can be involved in sculpting a malignant niche that supports LSC survival, as
depicted in Figure 1. Nevertheless, to make this point clearer for readers, we have added lines 291-318 (page 7)
to further underscore the notion that multiple interfaces are responsible for establishing the overall malignant
niche and sustaining cell-extrinsic signaling conducive to LSC persistence.
8) Moreover, the niche is supposedly dynamic and was both shown to be remodeled in some instances of
leukemia while being even an initiator of transformation in other cases (e.g. Wnt-overactivation, Dicer1 defects).
Both aspects have to be covered by the review.
We appreciate the reviewer’s insightful comment and have made appropriate additions in lines 291-318 (page 7)
to reflect the abundance of interactors and reciprocity of molecular aberrations in creating the leukemic niche by
the niche itself or by leukemic blasts or LSCs.
9) While being well structured in general, the section on Cell intrinsic signaling starts with a section of doubtful
structure. It is recommended to first approach the different ways how LSCs may be generated (from HSC or by
B C C A N C E R R E S E A R C H
675 West 10th Avenue Tel: 604.675.8000
Vancouver, BC, Canada V5Z 1L3 Fax: 604.877.0712
www.bccancer.bc.ca
dedifferentiation) and then go into the concrete details. And while it is of course hard to be discrete on such
topics, this first section jumps between signaling and transcriptomics. This also applies to epigenetic changes,
which are just another way to regulate the transcriptome and subsequent events and here the authors include
epi-transcriptomics. It is worth to consider a more general section on LSC induction and then move “naturally”
from epigenomics to transcriptomics, epi-transcriptomics, proteomocs and last the metabolism.
We thank the reviewer for excellent comments. We have now incorporated discussions on the generation of LSCs
from HSCs via pre-LSCs in lines 477-486 (page 12) and also acknowledged the generation of LSCs by dedifferentiation, particularly via the suppression of pioneer factor PU.1, in lines 515-519. Furthermore, we
acknowledge and resonate with the reviewer’s comment that it is difficult to be discrete when addressing
transcriptome without going into intracellular signaling mediated by aberrant/oncogenic transcription factors as
transcription factor signaling is fundamental to the overall transcriptomic dynamics. To resolve this, we have
modified the titles of the sub-sections to accurately re-capitulate the theme conveyed therein. For instance, we
have modified the original section 3.1.1 title from “Transcriptome of AML LSCs” to the current section 3.2.1 title
“Transcriptome and Transcription Factor Signaling of AML LSCs.” By the same token, we have also modified the
original section 3.1.2 title from “Epigenome of AML LSCs” to the current section 3.2.2 title “Epi-transcriptome and
Epigenome of AML LSCs,” and likewise for CML LSCs. This way, the titles not only reflect the intertwined nature
among these molecular systems but also more accurately inform the readers of the content discussed within the
subsections.
10) Why is CML treated separately? With the current title and topic, CML should be integrated.
We appreciate the reviewer’s feedback. The manuscript approached AML and CML separately but with
comparable depth and length overall, starting from surface antigens to cell-extrinsic and then to cell-intrinsic
signaling. After the title has been modified to “Properties of Leukemic Stem Cells in Regulating Drug Resistance
in Acute and Chronic Myeloid Leukemias” to better reflect the scope of the review, we deem that our discussions
of CML are of relevance.
11) Consider to re-arrange therapies to their respective topics. E.g. surface antigen directed therapies could
follow the section on immune-phenotype. To this reviewer it makes more sense to present the therapies with the
biology instead of taking all therapies in sum as an individual section. Please also note that the section of small
molecule inihibitors is rather limited, yet it would need an individual review to cover all potential options
presented today and should not be extended but also included where the biology is presented (which is already
the case with some strategies like Bcl2-inhibition). The same applies for combinatorial therapies.
We thank the reviewer for this thoughtful feedback.
We appreciate and acknowledge that there are abundant small molecule inhibitors available to specifically target
AML and CML LSCs; however, this subsection (section 4.2) is meant to provide select examples of the diverse
biological processes of LSCs, such as their epigenome and particular kinase signaling aspects that can be
targeted by small molecule inhibitors. After reviewing section 4.2, we realized that we can add another small
molecule inhibitor, Dynole 34-2, as a prime example of targeting leukemic niche signaling to supplement our
discussion of the section.
Since appreciating LSC-targeted therapies requires integrated understanding of molecular alterations in and out
of LSCs in the first place, and that writing a section that centres upon multiple themes, such as molecular
B C C A N C E R R E S E A R C H
675 West 10th Avenue Tel: 604.675.8000
Vancouver, BC, Canada V5Z 1L3 Fax: 604.877.0712
www.bccancer.bc.ca
pathologies and current therapeutics, may be expansive and potentially poses the risk of overwhelming readers,
we would like to keep the therapy section focused and placed at the end of the review, if possible.
Reviewer 2 Report
The paper of Zhai et al. is a very comprehensive and complete review of leukemic stem cells and their ability to persist in bone marrow of AML and CML patients and to give raise to persistent and/or relapsing disease.
The review article is very complete and clearly written. The senior author Dr Jiang made valuable contributions in the field.
Minor points:
I think mechanisms of resistance emerging from LSCs to approved drugs for treatment of AML including anthracyclines, aracytine, hypomethylating agents, IDH and FLT3 inhibitors should be more clearly discussed and highlighted in a table.
Author Response
The paper of Zhai et al. is a very comprehensive and complete review of leukemic stem cells and their ability to
persist in bone marrow of AML and CML patients and to give raise to persistent and/or relapsing disease.
The review article is very complete and clearly written. The senior author Dr Jiang made valuable contributions in
the field.
We thank the reviewer for the positive appraisal of the manuscript and are delighted to learn that our review
article is comprehensive in nature. We are also humbled by and grateful for the many inspiring colleagues, whose
work has significantly advanced leukemia research.
Minor points:
1) I think mechanisms of resistance emerging from LSCs to approved drugs for treatment of AML including
anthracyclines, aracytine, hypomethylating agents, IDH and FLT3 inhibitors should be more clearly discussed and
highlighted in a table.
We thank the reviewer for this excellent suggestion. While we are fully aware of the diversity of resistance
mechanisms emerging from LSCs, we have presented a few exemplary LSC resistance mechanisms to clinically
approved drugs indicated for AML in Table 2, titled “Exemplary LSC Resistance Mechanisms to Clinically
Approved AML Therapies.”